# The Neural Covariance SDE: Shaped Infinite Depth-and-Width Networks at Initialization

**Mufan (Bill) Li**
University of Toronto,
Vector Institute

**Mihai Nica**
University of Guelph,
Vector Institute

**Daniel M. Roy**
University of Toronto,
Vector Institute

## Abstract

The logit outputs of a feedforward neural network at initialization are conditionally Gaussian, given a random covariance matrix defined by the penultimate layer. In this work, we study the distribution of this random matrix. Recent work has shown that shaping the activation function as network depth grows large is necessary for this covariance matrix to be non-degenerate. However, the current infinite-width-style understanding of this shaping method is unsatisfactory for large depth: infinite-width analyses ignore the microscopic fluctuations from layer to layer, but these fluctuations accumulate over many layers.

To overcome this shortcoming, we study the random covariance matrix in the shaped infinite-depth-and-width limit. We identify the precise scaling of the activation function necessary to arrive at a non-trivial limit, and show that the random covariance matrix is governed by a stochastic differential equation (SDE) that we call the Neural Covariance SDE. Using simulations, we show that the SDE closely matches the distribution of the random covariance matrix of finite networks. Additionally, we recover an if-and-only-if condition for exploding and vanishing norms of large shaped networks based on the activation function.

## 1 Introduction

Of the many milestones in deep learning theory, the precise characterization of the infinite-width limit of neural networks at initialization as a Gaussian process with a non-random covariance matrix [1, 2] was a turning point. The so-called Neural Network Gaussian process (NNGP) theory laid the mathematical foundation to study various limiting training dynamics under gradient descent [3–12]. The Neural Tangent Kernel (NTK) limit formed the foundation for a rush of theoretical work, including advances in our understanding of generalization for wide networks [13–15]. Besides the NTK limit, the infinite-width mean-field limit was developed [16–19], where the different parameterization demonstrates benefits for feature learning and hyperparameter tuning [20–22].

Fundamentally, the infinite-width paradigm derives results from the assumption that the depth of the network is held *fixed* while the widths of all layers grow to infinity. Unfortunately, this assumption can be problematic for modeling real-world networks, as the microscopic fluctuations from layer to layer are neglected in this limit (see Figure 1). In particular, infinite-width predictions are shown to be poor approximations of real networks unless the depth is much less than the width [23, 24].

Impressive achievements of deep networks with billions of parameters crystallize the importance of understanding extremely large, deep neural networks (DNNs). An alternative to the infinite-width paradigm is the infinite-depth-and-width paradigm. In this setting, both the network depth $d$ and the width $n$ of each layer are simultaneously scaled to infinity, while their relative ratio $d/n$ remains fixed [23, 25–29]. Recent work also explores using $d/n$ as an effective perturbation parameter [30–

---

Correspondence: `mufan.li@mail.utoronto.ca`; `nicam@uoguelph.ca`; `daniel.roy@utoronto.ca`.

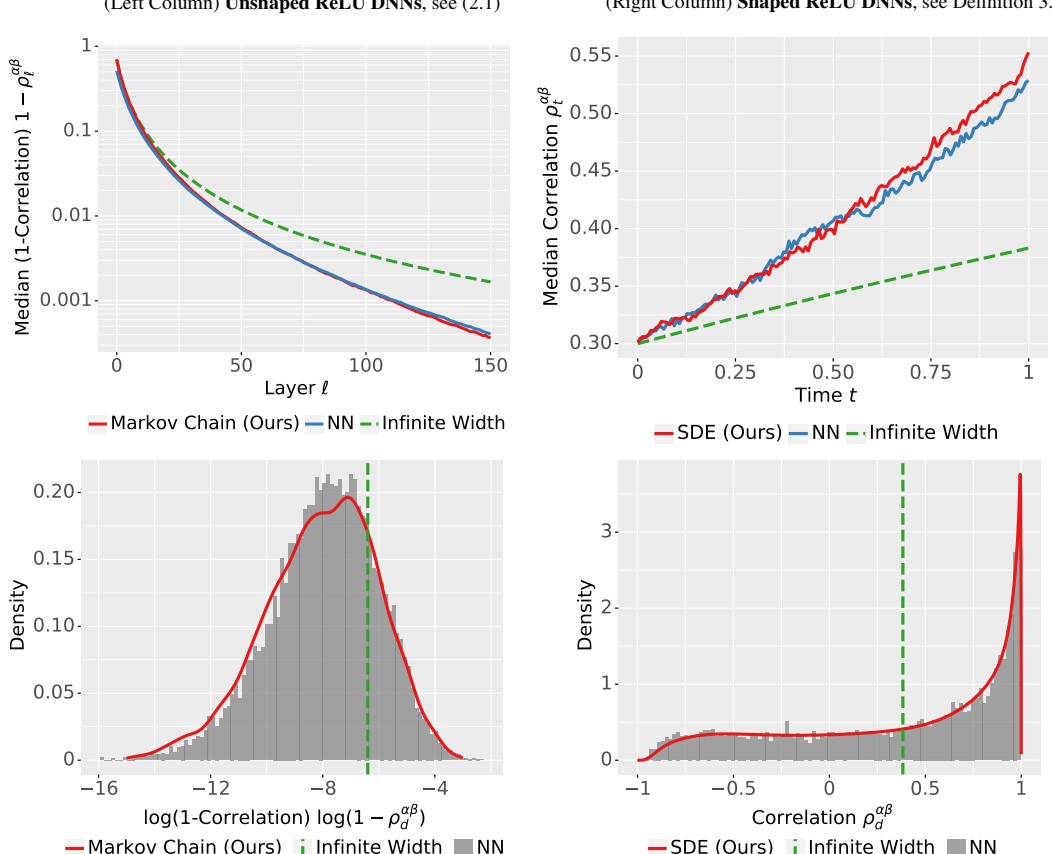

Figure 1: Simulations of correlation $\rho_\ell^{\alpha\beta} = \frac{\langle \varphi_\ell^\alpha, \varphi_\ell^\beta \rangle}{|\varphi_\ell^\alpha||\varphi_\ell^\beta|}$ between post-activation vectors in ReLU networks, comparing *finite NNs* vs. our theoretical predictions vs. infinite-width paradigm. **Left Column:** $\rho_\ell^{\alpha\beta}$ vs. our Markov chain (2.10) vs. infinite-width update $\rho_{\ell+1} = cK_1(\rho_\ell)$ (see (2.10) and note the log scale and $1 - \rho$ here). **Right Column:** $\rho_{\lfloor tn \rfloor}^{\alpha\beta}$ vs. our Neural Covariance SDE vs. ODE $d\rho_t = \nu(\rho_t) dt$ (see Theorem 3.3). **Top Row:** *Median $\rho$* as a function of layer. **Bottom Row:** *Full distribution* at final layer $\ell = d$. **Simulation details:** $n = d = 150, \rho_0 = 0.3, 2^{13}$ samples for each. In right column: $c_+ = 0, c_- = -1$, DE step size 1e$-2$. Densities from kernel density estimation.

33] or to study concentration bounds in terms of $d/n$ [5, 34]. This limit has the distinct advantage of being incredibly accurate at predicting the output distribution for finite size networks at initialization [27] — a significant improvement over the NNGP theory. Furthermore, it has also been shown that there is feature learning in this limit [23], in contrast to the linear regime of infinite-width limits [8]. Considering the mathematical success of the NNGP techniques, the infinite-depth-and-width limit hints at the possibility of developing an accurate theory for training and generalization.

An immediate issue of the infinite-depth limit is that this limit predicts that network output becomes degenerate as depth increases: on initialization the network becomes a constant function sending all inputs to the same (random) output [35, 36, 33]. While degenerate outputs are not necessarily an issue in theory, it poses a more serious problem in practice: degenerate correlations imply a "sharp" input–output Jacobian, and therefore exploding gradients [37, 25]. Intuitively, the output is not very sensitive to changes in the input, hence the gradient must be very large in the earlier layers.

A promising new attack on this problem is to modify the activation function ("shaping") to reduce to the effect of degeneracy [38, 39]. In this prior work, extensive experiments show that shaping the activation significantly improves training speed *without the need for normalization layers*. This method has been proven effective for problems as large as standard ResNets on ImageNet data. The authors designed several criteria including reducing estimated output correlation, and numerically

| Notation | Description | Notation | Description | Table 1: Notation |
|---|---|---|---|---|
| $n_{\text{in}} \in \mathbb{N}$ | Input dimension | $n_{\text{out}} \in \mathbb{N}$ | Output dimension | |
| $n \in \mathbb{N}$ | Hidden layer width | $d \in \mathbb{N}$ | Number of hidden layers (depth) | |
| $\varphi(\cdot)$ | Base activation | $\varphi_s(\cdot)$ | Shaped activation | |
| $x^\alpha \in \mathbb{R}^{n_{\text{in}}}$ | Input for $1 \leq \alpha \leq m$ | $W_0 \in \mathbb{R}^{n_{\text{in}} \times n}$ | Weight matrix at layer 0 | |
| $z_{\text{out}}^\alpha \in \mathbb{R}^{n_{\text{out}}}$ | Network output | $W_{\text{out}} \in \mathbb{R}^{n \times n_{\text{out}}}$ | Weight matrix at final layer | |
| $z_\ell^\alpha \in \mathbb{R}^n$ | Neurons (pre-activation) for layer $1 \leq \ell \leq d$ | $W_\ell \in \mathbb{R}^{n \times n}$ | Weight matrix at layer $1 \leq \ell \leq d$ **All weights initialized iid $\sim \mathcal{N}(0,1)$** | |
| $\varphi_\ell^\alpha \in \mathbb{R}^n$ | Neurons (post-activation) for layer $1 \leq \ell \leq d$ | $c \in \mathbb{R}$ | Normalizing constant $c := \left( \mathbb{E}\, \varphi(g)^2 \right)^{-1}$ for $g \sim \mathcal{N}(0,1)$ | |

optimized the shape of activation functions for improved training results. However, their deterministic estimation of output correlation using the infinite-width limit leads to a poor approximation of real networks, as the additional randomness has both non-zero mean and heavy skew (see Figure 1 right column). Furthermore, numerically searching for the activation shape obscures the picture on how shaping should depend on the network depth and width.

In this paper, we address these problems by providing a precise theory of shaped infinite-depth-and-width networks, extending both the NNGP theories and the activation shaping techniques. In particular, we prescribe an exact scaling of the activation function shape as a function of network width $n$ that leads to a non-trivial nonlinear limit. By keeping track of microscopic $O(n^{-1/2})$ random fluctuations in each layer of the network, we show that the cumulative effect is described by a stochastic differential equation (SDE) in the limit. In contrast to existing infinite-width theory, we are able to characterize the random distribution of the output covariance, which matches closely to simulations of real networks. In a similar spirit to how the NNGP theory laid the foundation for studying training and generalization in the infinite-width limit, we also see this work as building the mathematical tools for an infinite-depth-and-width theory of training and generalization.

## 1.1 Contributions

Similar to the NNGP approach, we use the fact that the output is Gaussian conditional on the penultimate layer. However, unlike in the infinite-width paradigm, the covariance matrix is no longer deterministic in the infinite-depth-and-width limit. Our focus in this paper is to study this random covariance matrix. Our main contributions are as follows:

1. We introduce the tool of stochastic $\sqrt{n}$-expansions and convergence to SDEs for analyzing the distribution of covariances in DNNs.
2. For *unshaped* ReLU-like activations, we show that the norm of each layer evolves according to geometric Brownian motion and correlations evolve according to a discrete Markov process. See left column of Figure 1 and Section 2.
3. For both ReLU-like and a large class of smooth activation functions, we derive the Neural Covariance SDE characterizing the distribution of the shaped infinite-depth-and-width limit. See right column of Figure 1 and Section 3.
4. We show our prescribed shape scaling is exact, as other rates of scaling leads to either degenerate or linear network limits. See Proposition 3.4 and Proposition 3.10.
5. For smooth activations, we derive an if-and-only-if condition for exploding/vanishing norms based on properties of the activation function. See Proposition 3.7 and Section 4.
6. We provide simulations to verify theoretical predictions and help interpret properties of real DNNs. See Figures 1 and 2 and supplemental simulations in Appendix F.

## 2  Limits for Unshaped ReLU-Like Activations

Using the **notation in Table 1**, the output of a fully connected feedforward network with $d$ hidden layers of width $n$ on input $x^\alpha$ is defined by vectors of **pre-activations** $z_\ell^\alpha$ and **post-activations** $\varphi_\ell^\alpha$:

$$z_1^\alpha := \frac{1}{\sqrt{n_{\text{in}}}} W_0 x^\alpha, \quad \varphi_\ell^\alpha := \varphi(z_\ell^\alpha), \quad z_{\ell+1}^\alpha := \sqrt{\frac{c}{n}} W_\ell \varphi_\ell^\alpha, \quad z_{\text{out}}^\alpha := \sqrt{\frac{c}{n}} W_{\text{out}} \varphi_d^\alpha. \tag{2.1}$$

Note that factors of $\sqrt{cn^{-1}}$ are equivalent to intializing according to the so-called He initialization [40]. We use Greek indices $\alpha, \beta, \dots$ to denote multiple different inputs. Note that while our results are all stated for fixed width $n$ in each layer, they can be generalized to layer width $n_\ell$ in the limit where all $n_\ell \to \infty$ with $\sum_{\ell=1}^d n_\ell^{-1}$ replacing the role of the depth-to-width ratio $d/n$ [25].

In this section, we analyze **ReLU-like** activations by which we mean activations which are linear on the negative and positive numbers given respectively by two slopes $s_+$ and $s_-$:

$$\varphi(x) := s_+ \max(x, 0) + s_- \min(x, 0) = s_+ \varphi_{\text{ReLU}}(x) - s_- \varphi_{\text{ReLU}}(-x). \tag{2.2}$$

These are precisely the **positive homogeneous** functions: $\varphi(ax) = |a|\,\varphi(x)\,\forall x, a \in \mathbb{R}$.

## 2.1 SDE Limits of Markov Chains

We briefly review the main type of SDE convergence principle used in our main results (see Proposition A.6 for a more precise version). Let $X_t$, $t \in \mathbb{R}^+$, be a continuous time diffusion process obeying an SDE with drift $b$ and variance $\sigma^2$ as given in (2.3). Suppose that for each $n \in \mathbb{N}$, $Y_\ell^n$ is a discrete time Markov chain $\ell \in \mathbb{N}$ whose increments obey (2.3) in terms of the same functions $b, \sigma^2$:

$$dX_t = b(X_t)\,dt + \sigma(X_t)\,dB_t, \qquad Y_{\ell+1}^n - Y_\ell^n = b(Y_\ell^n)\frac{1}{n} + \sigma(Y_\ell^n)\frac{\xi_\ell}{\sqrt{n}} + O(n^{-3/2}), \tag{2.3}$$

where $\xi_\ell$ are independent variables with $\mathbb{E}(\xi_\ell) = 0, \mathbf{Var}(\xi_\ell) = 1$. With this setup, under technical conditions described precisely in Appendix A, we have convergence of $Y_\ell$ at $\ell = \lfloor tn \rfloor$ to $X_t$, or more precisely: with $X_t^n := Y_{\lfloor tn \rfloor}^n$ we have $X^n \to X$ as $n \to \infty$ in the Skorohod topology. In our applications, $n$ is always the width (i.e., number of neurons in each layer) which may appear implicitly and $\ell$ is always the layer number.

## 2.2 A Simple SDE: Geometric Brownian Motion Describes $|\varphi_\ell^\alpha|^2$

To motivate our approach of SDE limits, we illustrate the method using the example of the squared norm of the $\ell$-th layer, $|\varphi_\ell^\alpha|^2$, where we recall $\varphi_\ell^\alpha = \varphi(z_\ell^\alpha)$. For a single fixed input $x^\alpha$ and a ReLU-like activation $\varphi$, the norm of the post-activation neurons $|\varphi_\ell^\alpha|^2$ forms a Markov chain in the layer number $\ell$. We use the fact that a matrix with iid Gaussian entries applied to any unit vector gives a Gaussian vector of iid $\mathcal{N}(0,1)$ entries. Hence, in each layer, we can define the Gaussian vector $g^\alpha$ as follows, and use (2.1) with the positive homogeneity of $\varphi$ to write the Markov chain update rule:

$$\left|\varphi_{\ell+1}^\alpha\right|^2 = |\varphi_\ell^\alpha|^2 \frac{1}{n}\sum_{i=1}^n c\varphi(g_i^\alpha)^2, \quad \text{where } g^\alpha := W_\ell \frac{\varphi_\ell^\alpha}{|\varphi_\ell^\alpha|} \stackrel{d}{=} \mathcal{N}(0, I_n). \tag{2.4}$$

At this point, the infinite-width approach applies the law of large numbers (LLN) to conclude $\lim_{n\to\infty} \left|\varphi_{\ell+1}^\alpha\right|^2 = |\varphi_\ell^\alpha|^2 \mathbb{E}[c\varphi^2(g)] = |\varphi_\ell^\alpha|^2 \cdot 1$ a.s. by definition of $c$. However, the LLN cannot be applied when depth $d$ is diverging with $n$, as the cumulative effect of the fluctuations over $d$ layers does not vanish! Instead, we keep track of the $O(1/\sqrt{n})$ fluctuations in each layer by introducing the zero mean finite variance random variable $R_\ell^{\alpha\alpha} := \frac{1}{\sqrt{n}}\sum_{i=1}^n \left(c\varphi(g_i^\alpha)^2 - 1\right)$. This allows us to rewrite this Markov chain update rule as

$$\left|\varphi_{\ell+1}^\alpha\right|^2 = |\varphi_\ell^\alpha|^2 \left(1 + \frac{1}{\sqrt{n}}R_\ell^{\alpha\alpha}\right), \tag{2.5}$$

which allows us to see that the Markov chain $Y_\ell^n = \frac{c}{n}|\varphi_\ell^\alpha|^2$ is now in the form of (2.3) with $Y_0^n = \frac{1}{n_{\text{in}}}|x^\alpha|^2$, $b(Y) \equiv 0, \sigma^2(Y) = \mathbf{Var}(R_\ell^{\alpha\alpha})Y^2 = \mathbf{Var}(c\varphi(g)^2)Y^2$. Consequently, we have that the squared norm Markov chain converges to a geometric Brownian motion $dX_t = \sigma X_t dB_t$, or more precisely

$$\lim_{n\to\infty} \frac{c}{n}\left|\varphi_{\lfloor tn \rfloor}^\alpha\right|^2 = X_t \stackrel{d}{=} e^{\mathcal{N}(-\frac{\sigma^2}{2}t, \sigma^2 t)}, \tag{2.6}$$

where the convergence is in the Skorohod topology (see Appendix A). When $\varphi$ is the ReLU function ($s_+ = 1, s_- = 0$), we have $c = 2$ and $\sigma^2 = 5$, which recovers known results in [25, 27–29]. We remark again this simple Markov chain example illustrates the main technique we use in later sections to establish SDE convergence for shaped networks in Section 3.

## 2.3 Non-SDE Markov Chains: the Gram Matrix $\left\langle \varphi_\ell^\alpha, \varphi_\ell^\beta \right\rangle$ and Correlation $\rho_\ell^{\alpha\beta}$

We can generalize Section 2.2 to a collection of $m$ inputs $\{x^\alpha\}_{\alpha=1}^m$ by looking at the entire Gram matrix $[\langle \varphi_\ell^\alpha, \varphi_\ell^\beta \rangle]_{\alpha,\beta=1}^m$, where we again recall $\varphi_\ell^\alpha = \varphi(z_\ell^\alpha)$. We note that the convergence of Markov chains to SDEs in (2.3) can be generalized to $Y_\ell^n \in \mathbb{R}^N$ by considering $\mathbf{Cov}(\xi_\ell) = I_N$, $b : \mathbb{R}^N \to \mathbb{R}^N$, and $\sigma : \mathbb{R}^N \to \mathbb{R}^{N \times N}$. The Gram matrix is of particular interest because the neurons in any layer are conditionally Gaussian **when conditioned on the previous layer**, with covariance matrix proportional to the Gram matrix:

$$[z_{\ell+1}^\alpha]_{\alpha=1}^m \big| \mathcal{F}_\ell \overset{d}{=} \mathcal{N}\left(0, \frac{c}{n}[\langle \varphi_\ell^\alpha, \varphi_\ell^\beta \rangle]_{\alpha,\beta=1}^m \otimes I_n\right),$$
$$[z_{\text{out}}^\alpha]_{\alpha=1}^m \big| \mathcal{F}_d \overset{d}{=} \mathcal{N}\left(0, \frac{c}{n}[\langle \varphi_d^\alpha, \varphi_d^\beta \rangle]_{\alpha,\beta=1}^m \otimes I_{n_{\text{out}}}\right),$$

(2.7)

where $\mathcal{F}_\ell$ denotes the sigma-algebra generated by the $\ell$-th layer $[z_\ell^\alpha]_{\alpha=1}^m$, and $\otimes$ denotes the Kronecker product (here indicating conditionally independent entries in each vector). With this property in mind, we will introduce $\mathbb{E}_\ell[\cdot] := \mathbb{E}[\cdot | \mathcal{F}_\ell]$ to denote the conditional expectation, and $\mathbf{Var}_\ell(\cdot), \mathbf{Cov}_\ell(\cdot)$ similarly to denote the conditional variance and covariance. If we define $g^\alpha$ as in (2.4), we see that the $g^\alpha$ are all marginally $\mathcal{N}(0, I_n)$. Similar to (2.4), we can write the update rule for the $\alpha, \beta$-entry of the Gram matrix:

$$\langle \varphi_{\ell+1}^\alpha, \varphi_{\ell+1}^\beta \rangle = |\varphi_\ell^\alpha||\varphi_\ell^\beta| \frac{1}{n} \sum_{i=1}^n c\varphi(g_i^\alpha)\varphi(g_i^\beta),$$

(2.8)

Just as we did in (2.5), we can define $R_\ell^{\alpha\beta} := \frac{1}{\sqrt{n}} \sum_{i=1}^n c\varphi(g_i^\alpha)\varphi(g_i^\beta) - \mathbb{E}_\ell[c\varphi(g_i^\alpha)\varphi(g_i^\beta)]$ and write

$$\langle \varphi_{\ell+1}^\alpha, \varphi_{\ell+1}^\beta \rangle = |\varphi_\ell^\alpha||\varphi_\ell^\beta| \left( \mathbb{E}_\ell\left[ c\varphi(g_i^\alpha)\varphi(g_i^\beta) \right] + \frac{1}{\sqrt{n}} R_\ell^{\alpha\beta} \right),$$

(2.9)

where $R_\ell^{\alpha\beta}$ are mean zero with covariance $\mathbf{Cov}_\ell[R_\ell^{\alpha\beta}, R_\ell^{\gamma\delta}] = \mathbf{Cov}_\ell[c\varphi(g^\alpha)\varphi(g^\beta), c\varphi(g^\gamma)\varphi(g^\delta)]$. (By the Central Limit Theorem, $R_\ell^{\alpha\beta}$ will be approximately Gaussian for large $n$.)

However, unlike the simple single-data-point case from Section 2.2, **we do not have convergence to a continuous time SDE.** This is because the differences $\langle \varphi_{\ell+1}^\alpha, \varphi_{\ell+1}^\beta \rangle - \langle \varphi_\ell^\alpha, \varphi_\ell^\beta \rangle \not\to 0$ as $n \to \infty$. Instead, (2.9) is a discrete recursion update with additive noise of the form $Y_{\ell+1}^n = f(Y_\ell^n) + \frac{1}{\sqrt{n}}\xi$ for some function $f$, and consequently $Y_{\ell+1}^n - Y_\ell^n$ does not vanish as $n \to \infty$.

For a clarifying example, we can consider the one-dimensional Markov chain of hidden layer correlations. More precisely, we can define $\rho_\ell^{\alpha\beta} = \langle \varphi_\ell^\alpha, \varphi_\ell^\beta \rangle / |\varphi_\ell^\alpha||\varphi_\ell^\beta|$, which we observe can be extracted from the entries of the Gram matrix. In fact, we can write down an approximate recursion update for $\rho_\ell^{\alpha\beta}$ (see Appendix B and Proposition B.8 for details):

$$\rho_{\ell+1}^{\alpha\beta} \approx cK_1(\rho_\ell^{\alpha\beta}) + \frac{1}{n}\mu_{\text{ReLU}}(\rho_\ell^{\alpha\beta}) + \frac{\xi_\ell}{\sqrt{n}}\sigma_{\text{ReLU}}(\rho_\ell^{\alpha\beta}), \quad \rho_0^{\alpha\beta} = \frac{\langle x^\alpha, x^\beta \rangle}{n_{\text{in}}},$$

(2.10)

where $K_1(\rho) := \mathbb{E}[\varphi(g)\varphi(g\rho + w\sqrt{1-\rho^2})]$ for $g, w$ iid $\mathcal{N}(0, 1)$ random variables, and $\xi_\ell$ are iid $N(0, 1)$. For the ReLU case, $c = 2$ and $cK_1(\rho) = (\sqrt{1-\rho^2} + \rho \arccos(-\rho))/\pi$ was first calculated in [41]. In fact, we can observe that as $n \to \infty$, $\rho_{\lfloor tn \rfloor}^{\alpha\beta}$ converges to the fixed point of $cK_1(\cdot)$ at $\rho = 1$ for all $t > 0$. **We note this limiting behaviour cannot be described by an SDE**, as the solution must jump from the initial condition to the fixed point at $t = 0$.

Despite not having an SDE limit, we observe that the approximate Markov chain (2.10) already provides a much better approximation to finite size networks compared to the infinite-width theory (see left column of Figure 1). This is because the infinite-width approach discards the terms in (2.10) that vanish as $n \to \infty$ and consider only the update $\rho_{\ell+1}^{\alpha\beta} = cK_1(\rho_\ell^{\alpha\beta})$. Analysis of this deterministic equation leads to the prediction that $\rho_\ell^{\alpha\beta} = 1 - O(\ell^{-2})$ for $\ell \gg 1$ (see (4.8) in [33] and a new bound in Appendix E).

Furthermore, we observe that in this case, the microscopic $O(n^{-1})$ and $O(n^{-1/2})$ terms in (2.10) accumulate to macroscopic differences! For the examples in Figure 1, we see their net effect is that

$\rho_\ell^{\alpha\beta} \to 1$ faster than the infinite-width prediction. Heuristically, the reason for this discrepancy is due to $\sigma_{\text{ReLU}}(\rho) \to 0$ as $\rho \to 1$. This means that the randomness can push $\rho_\ell^{\alpha\beta}$ closer to 1, but becomes "trapped" when $\rho_\ell^{\alpha\beta}$ is close to 1 because $\sigma_{\text{ReLU}}$ is so small here. In the next section, we will see that we are just one step away from achieving limiting SDEs.

# 3  Neural Covariance SDEs: Shaped Infinite-Depth-and-Width Limit

In this section, we follow the ideas of [38, 39] to *reshape* the activation function $\varphi$. Reshaping means to replace the base activation function $\varphi$ in (2.1) with $\varphi_s$ that depends on width $n$. We will also replace the normalizing constant $c = \left( \mathbb{E}\, \varphi_s(g)^2 \right)^{-1}$ for $g \sim \mathcal{N}(0, 1)$. Specifically, we will choose $\varphi_s$ to depend on $n$ such that in the limit as $n \to \infty$, we have that $\varphi$ is approximately an identity function, $\varphi_s \to \text{Id}$. Recalling from (2.7) that the output is conditionally Gaussian with covariance determined by the Gram matrix $[\langle \varphi_\ell^\alpha, \varphi_\ell^\beta \rangle]_{\alpha,\beta=1}^m$, therefore we recover a complete characterization by describing the random covariance matrix.

## 3.1  Neural Covariance SDE for Shaped ReLU-Like Activations

**Definition 3.1.** *We shape the ReLU-like activation $\varphi_s(x) := s_+ \max(x, 0) + s_- \min(x, 0)$, by setting the slopes to depend on $n$ according to $s_\pm := 1 + \frac{c_\pm}{\sqrt{n}}$ for some given constants $c_+, c_- \in \mathbb{R}$. We will also set $c = \left( \mathbb{E}\, \varphi_s(g)^2 \right)^{-1}$ for $g \sim \mathcal{N}(0, 1)$.*

We will show that with shaping of Definition 3.1, one gets non-trivial SDEs that describe the covariance (Theorem 3.2) and correlations (Theorem 3.3) of the network. The precise scaling is shown to be the critical scaling for a non-trivial limit in Proposition 3.4. All proofs for results in this section appear in Appendix C.

*Remark.* Note that in the statement of our theorems, we abuse notation and use the same letter to denote the pre-limit Markov chain and the limiting SDE. For example, in Theorem 3.2 we use $V_\ell$ for the covariance at layer $\ell$ and $V_t$ to denote the limiting SDE at time $t$.

**Theorem 3.2** (Covariance SDE, ReLU). *Let $V_\ell^{\alpha\beta} := \frac{c}{n} \langle \varphi_\ell^\alpha, \varphi_\ell^\beta \rangle$, and define $V_\ell := [V_\ell^{\alpha\beta}]_{1 \le \alpha \le \beta = m}$ to be the upper triangular entries thought of as a vector in $\mathbb{R}^{m(m+1)/2}$. Then, with $s_\pm = 1 + \frac{c_\pm}{\sqrt{n}}$ as in Definition 3.1, in the limit as $n \to \infty, \frac{d}{n} \to T$, the interpolated process $V_{\lfloor tn \rfloor}$ converges in distribution in the Skorohod topology of $D_{\mathbb{R}_+, \mathbb{R}^{m(m+1)/2}}$ to the solution of the SDE*

$$dV_t = b(V_t)\, dt + \Sigma(V_t)^{1/2}\, dB_t\,, \quad V_0 = \left[ \frac{1}{n_{in}} \langle x^\alpha, x^\beta \rangle \right]_{1 \le \alpha \le \beta \le m}, \tag{3.1}$$

*where $\nu(\rho) := \frac{(c_+ - c_-)^2}{2\pi} \left( \sqrt{1 - \rho^2} - \rho \arccos \rho \right), \rho_t^{\alpha\beta} := \frac{V_t^{\alpha\beta}}{\sqrt{V_t^{\alpha\alpha} V_t^{\beta\beta}}}$*

$$b(V_t) = \left[ \nu\left( \rho_t^{\alpha\beta} \right) \sqrt{V_t^{\alpha\alpha} V_t^{\beta\beta}} \right]_{1 \le \alpha \le \beta \le m}, \quad \text{and} \quad \Sigma(V_t) = \left[ V_t^{\alpha\gamma} V_t^{\beta\delta} + V_t^{\alpha\delta} V_t^{\beta\gamma} \right]_{\alpha \le \beta, \gamma \le \delta}. \tag{3.2}$$

*Furthermore, the output distribution can be described conditional on $V_T$ evaluated at final time $T$*

$$[z_{out}^\alpha]_{\alpha=1}^m \,|\, V_T \stackrel{d}{=} \mathcal{N}\left( 0, [V_T^{\alpha\beta}]_{\alpha,\beta=1}^m \right). \tag{3.3}$$

Here we remark that $\nu(1) = 0$, and therefore the drift component of diagonal entries $(V_t^{\alpha\alpha})$ are zero, as they are geometric Brownian motion. However, we emphasize that the $m$-point joint output distribution is *not* characterized by the marginal for each of the pairs, as the output $z_{\text{out}}^\alpha$ is *not* Gaussian. In particular, we observe the diffusion matrix entry corresponding to $V_t^{\alpha\beta}, V_t^{\gamma\delta}$ involves other processes $V_t^{\alpha\gamma}, V_t^{\beta\delta}, V_t^{\alpha\delta}, V_t^{\beta\gamma}$! *This implies that the Neural Covariance SDE limit cannot be described by a kernel, unlike stacking random features or NNGP.*

That being said, it is still instructive to study the marginal for a pair of data points. More specifically, it turns out in the generalized ReLU case, we can derive the marginal SDE for the correlation process.

**Theorem 3.3** (Correlation SDE, ReLU). *Let* $\rho_\ell^{\alpha\beta} := \frac{\langle \varphi_\ell^\alpha, \varphi_\ell^\beta \rangle}{|\varphi_\ell^\alpha| \, |\varphi_\ell^\beta|}$, *where* $\varphi_\ell^\alpha := \varphi_s(z_\ell^\alpha)$. *In the limit as* $n \to \infty$ *and* $s_\pm = 1 + \frac{c_\pm}{\sqrt{n}}$, *the interpolated process* $\rho_{\lfloor tn \rfloor}^{\alpha\beta}$ *converges in distribution to the solution of the following SDE in the Skorohod topology of* $D_{\mathbb{R}_+, \mathbb{R}}$

$$d\rho_t^{\alpha\beta} = \left[ \nu(\rho_t^{\alpha\beta}) + \mu(\rho_t^{\alpha\beta}) \right] dt + \sigma(\rho_t^{\alpha\beta}) \, dB_t \,, \quad \rho_0^{\alpha\beta} = \frac{\langle x^\alpha, x^\beta \rangle}{|x^\alpha| \, |x^\beta|} \,, \tag{3.4}$$

*where*

$$\nu(\rho) = \frac{(c_+ - c_-)^2}{2\pi} \left[ \sqrt{1 - \rho^2} - \arccos(\rho)\rho \right] \,, \quad \mu(\rho) = -\frac{1}{2}\rho(1 - \rho^2) \,, \quad \sigma(\rho) = 1 - \rho^2 \,. \tag{3.5}$$

To help interpret the SDE, we observe that $\mu$ and $\sigma$ are entirely independent of the activation function. In other words, these terms will be present in this limit even for linear networks. At the same time, $\nu$ describes the influence of the shaped activation function in this limit. [39] has derived a related ordinary differential equation (ODE) of $d\rho_t = \nu(\rho_t) \, dt$ in the sequential limit of $n \to \infty$ then $d \to \infty$, where the activation is shaped depending on depth. Here we also note that $\nu(\rho)$ is closely related to the $J_1$ function derived in [41]. See Appendix C.3 for the $m$-point joint version of the correlation SDE, and Appendix F for an empirical measure of convergence in the Kolmogorov–Smirnov distance.

One immediate consequence of the correlation SDE is that we can show the $n^{-1/2}$ scaling in Definition 3.1 is the only case where the limit is neither degenerate nor a linear network.

**Proposition 3.4** (Critical Exponent, ReLU). *Let* $\rho_\ell^{\alpha\beta} := \frac{\langle \varphi_\ell^\alpha, \varphi_\ell^\beta \rangle}{|\varphi_\ell^\alpha| \, |\varphi_\ell^\beta|}$, *where* $\varphi_\ell^\alpha := \varphi_s(z_\ell^\alpha)$. *Consider the limit* $n \to \infty$ *and* $s_\pm = 1 + \frac{c_\pm}{n^p}$ *for some* $p \geq 0$. *Then depending on the value of* $p$, *the interpolated process* $\rho_{\lfloor tn \rfloor}^{\alpha\beta}$ *converges in distribution w.r.t. the Skorohod topology of* $D_{\mathbb{R}_+, \mathbb{R}}$ *to*

(i) *the degenerate limit:* $\rho_t^{\alpha\beta} = 1$ *for all* $t > 0$, *if* $0 \leq p < \frac{1}{2}$, *and* $c_+ \neq c_-$,

(ii) *the critical limit: the SDE from Theorem 3.3, if* $p = \frac{1}{2}$,

(iii) *the linear network limit: if* $p > \frac{1}{2}$, *the following SDE, with* $\mu, \sigma$ *as defined in* (3.5),

$$d\rho_t^{\alpha\beta} = \mu(\rho_t^{\alpha\beta}) \, dt + \sigma(\rho_t^{\alpha\beta}) \, dB_t \,, \quad \rho_0^{\alpha\beta} = \frac{\langle x^\alpha, x^\beta \rangle}{|x^\alpha| \, |x^\beta|} \,. \tag{3.6}$$

Here we remark that the unshaped network case ($p = 0$) is contained by the above in case (i). At the same time, we observe that case (iii) is equivalent to the correlation SDE in Theorem 3.3 except with $\nu = 0$. In particular, we observe this limit is also reached when $c_+ = c_-$, which implies $\varphi_s(x) = s_+ x$ is linear, which is the reason we call this the linear network limit. Furthermore, without much additional work, the same argument also implies the joint covariance SDE also loses the drift component, i.e., $dV_t = \Sigma(V_t)^{1/2} \, dB_t$.

## 3.2 Neural Covariance SDE for Shaped Smooth Activations

In this section, we consider smooth activation functions and derive a similar covariance SDE. All the proofs for results in this section can be found in Appendix D.

**Assumption 3.5.** $\varphi \in C^4(\mathbb{R})$, $\varphi(0) = 0, \varphi'(0) = 1$, *and* $|\varphi^{(4)}(x)| \leq C(1 + |x|^p)$ *for some* $C, p > 0$.

We note that for any non-constant function $\sigma \in C^1(\mathbb{R})$ and $x_0 \in \mathbb{R}$ such that $\sigma'(x_0) \neq 0$, we can always define $\varphi(x) := \frac{\sigma(x + x_0) - \sigma(x_0)}{\sigma'(x_0)}$ such that it satisfies $\varphi(0) = 0, \varphi'(0) = 1$. The choice of $x_0$ will be discussed further in Section 4. The fourth derivative growth condition is used to control the Taylor remainder term in expectation, but any control over the remainder will suffice.

Following the ideas of [38], we consider the following shaping of a smooth activation function $\varphi$.

**Definition 3.6.** *For some constant* $a > 0$, *we set* $\varphi_s(x) := s\varphi\left(\frac{x}{s}\right)$ *with* $s = a\sqrt{n}$, *and* $c = \left( \mathbb{E} \, \varphi_s(g)^2 \right)^{-1}$ *for* $g \sim \mathcal{N}(0, 1)$.

Observe that in the limit $n \to \infty$, we will achieve that $\varphi_s \to \text{Id}$ as desired. We also observe that the shaping factor $s$ outside the activation cancels out with the next layer's $\frac{1}{s}$ factor, therefore it is equivalent shape the entire network. More precisely, if we view $z_{\text{out}}$ as an input-output map $f : \mathbb{R}^{n_{\text{in}}} \to \mathbb{R}^{n_{\text{out}}}$ of an unshaped network, then shaping the smooth activation functions is equivalent to the modification $sf\left(\frac{x}{s}\right)$.[1]

In this regime, we can similarly characterize the joint output distribution, *however the limiting SDEs are not always well behaved.* In particular, they can have finite time explosions as described by the Feller test for explosions [42, Theorem 5.5.29]. Here the SDE in Proposition 3.7 is exactly the $V_t^{\alpha\alpha}$ marginal of the Neural Covariance SDE, with the parameter $b$ determined by the activation function $\varphi$ and controls whether or not finite time explosions happen (see (4.1)).

**Proposition 3.7** (Finite Time Explosion). *Let $X_t \in \mathbb{R}_+$ be a solution to the following SDE*

$$dX_t = bX_t(X_t - 1)\,dt + \sqrt{2}X_t\,dB_t\,, \quad X_0 = x_0 > 0\,, b \in \mathbb{R}\,. \tag{3.7}$$

*Let $\tau^* = \sup_{M>0}\inf\{t : X_t \geq M \text{ or } X_t \leq M^{-1}\}$ be the explosion time, and we say $X_t$ has a finite time explosion if $\tau^* < \infty$. For this equation, $\mathbb{P}[\tau^* = \infty] = 1$ if and only if $b \leq 0$.*

Technically speaking, the main culprit behind finite time explosions is the non-Lipschitzness of the drift coefficient. This issue requires us to weaken the sense of convergence in this section; the ordinary convergence in the Skorohod topology is in general not true when the diffusion has finite time explosions. A weakened type of convergence is the best we can hope for. To this goal, we introduce the following definition.

**Definition 3.8.** *We say a sequence of processes $X^n$ **converge locally** to $X$ in the Skorohod topology if for any $r > 0$, we define the following stopping times*

$$\tau^n := \{t \geq 0 : |X_t^n| \geq r\}\,, \quad \tau := \{t \geq 0 : |X_t| \geq r\}\,, \tag{3.8}$$

*and we have that $X_{t\wedge\tau^n}^n$ converge to $X_{t\wedge\tau}$ in the Skorohod topology.*

This weakened sense of convergence essentially constrains the processes $X^n, X$ in a bounded set by adding an absorbing boundary condition. Not only do these stopping times rule out explosions, the drift coefficient is now also Lipschitz on a compact set. With this notion of convergence, we can now state a precise Neural Covariance SDE result for general smooth activation functions.

**Theorem 3.9** (Covariance SDE, Smooth). *Let $\varphi$ satisfy Assumption 3.5, $V_\ell^{\alpha\beta} := \frac{c}{n}\langle \varphi_\ell^\alpha, \varphi_\ell^\beta \rangle$ where $\varphi_\ell^\alpha = \varphi_s(z_\ell^\alpha)$, and define $V_\ell := [V_\ell^{\alpha\beta}]_{1\leq\alpha\leq\beta=m}$ to be the upper triangular entries thought of as a vector in $\mathbb{R}^{m(m+1)/2}$. Then, with $s = a\sqrt{n}$ as in Definition 3.6, in the limit as $n \to \infty, \frac{d}{n} \to T$, the interpolated process $V_{\lfloor tn \rfloor}$ converges locally in distribution to the solution of the following SDE in the Skorohod topology of $D_{\mathbb{R}_+, \mathbb{R}^{m(m+1)/2}}$*

$$dV_t = b(V_t)\,dt + \Sigma(V_t)^{1/2}\,dB_t\,, \quad V_0 = \left[\frac{1}{n_{in}}\langle x^\alpha, x^\beta \rangle\right]_{1\leq\alpha\leq\beta\leq m}, \tag{3.9}$$

*where $\Sigma(V_t)$ is the same as Theorem 3.2 and*

$$b^{\alpha\beta}(V_t) = \frac{\varphi''(0)^2}{4a^2}\left(V_t^{\alpha\alpha}V_t^{\beta\beta} + V_t^{\alpha\beta}(2V_t^{\alpha\beta} - 3)\right) + \frac{\varphi'''(0)}{2a^2}V_t^{\alpha\beta}(V_t^{\alpha\alpha} + V_t^{\beta\beta} - 2)\,. \tag{3.10}$$

*Furthermore, if $V_T$ is finite, then the output distribution can be described conditional on $V_T$ as*

$$[z_{out}^\alpha]_{\alpha=1}^m \,|V_T \overset{d}{=} \mathcal{N}\left(0, [V_T^{\alpha\beta}]_{\alpha,\beta=1}^m\right), \tag{3.11}$$

*and otherwise the distribution of $[z_{out}^\alpha]_{\alpha=1}^m$ is undefined.*

We also have a similar critical scaling result for general smooth activations.

**Proposition 3.10** (Critical Exponent, Smooth). *Let $\varphi$ satisfy Assumption 3.5, $V_\ell^{\alpha\beta} := \frac{c}{n}\langle \varphi_\ell^\alpha, \varphi_\ell^\beta \rangle$ where $\varphi_\ell^\alpha = \varphi_s(z_\ell^\alpha)$ with $s = an^p$ for some $p > 0$, and define $V_\ell := [V_\ell^{\alpha\beta}]_{1\leq\alpha\leq\beta=m}$ to be the upper triangular entries thought of as a vector. Then in the limit as $n \to \infty, \frac{d}{n} \to T$, the interpolated process $V_{\lfloor tn \rfloor}$ converges locally in distribution w.r.t. the Skorohod topology of $D_{\mathbb{R}_+, \mathbb{R}^{m(m+1)/2}}$ to $V$, which depending on the value of $p$ is*

---

[1]We want to thank Boris Hanin for observing this equivalent parameterization.

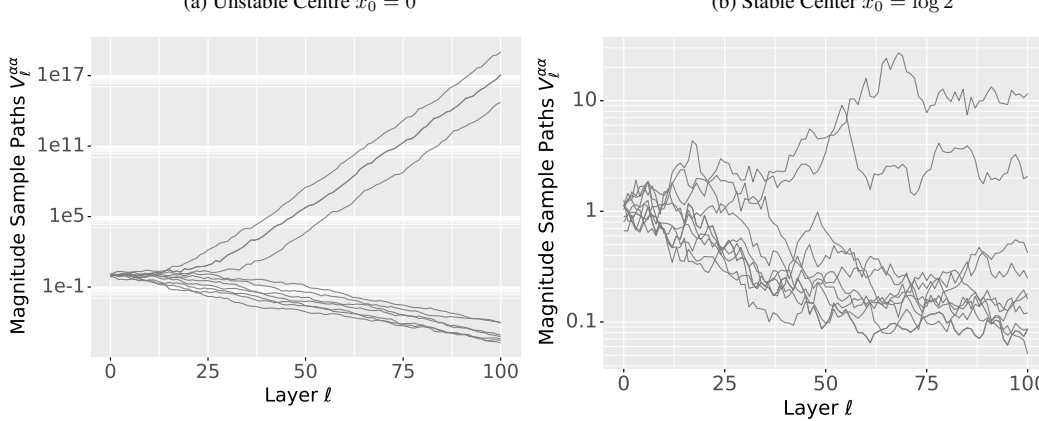

Figure 2: Simulation of 10 shaped softplus networks as in Example 4.2 with $n = d = 100, a = 1, V_0^{\alpha\alpha} = \frac{1}{n_{in}}|x^\alpha|^2 = 1$ centred at two different values. "Stable" here means the Neural Covariance SDE is guaranteed not to have finite time explosions; unstable networks can explode on initialization!

(i) *the degenerate limit: if* $0 < p < \frac{1}{2}$

$$\begin{cases} V_t^{\alpha\alpha} = 0 \text{ or } \infty, & \text{if } \frac{3}{4}\varphi''(0)^2 + \varphi'''(0) > 0 \text{ and } V_0^{\alpha\alpha} \neq 0, \\ V_t^{\alpha\beta} = const., & \text{if } \frac{3}{4}\varphi''(0)^2 + \varphi'''(0) \leq 0, \end{cases} \quad (3.12)$$

*for all* $t > 0$ *and* $1 \leq \alpha \leq \beta \leq m$,

(ii) *the critical limit: the solution of the SDE from Theorem 3.9, if* $p = \frac{1}{2}$,

(iii) *the linear network limit: the stopped solution to the SDE* $dV_t = \Sigma(V_t)\, dB_t$ *with coefficient* $\Sigma$ *defined in Theorem 3.3, if* $p > \frac{1}{2}$.

Here we observe that in case (i) when $\frac{3}{4}\varphi''(0)^2 + \varphi'''(0) \leq 0$, we also have a constant (in time) correlation $\rho_t^{\alpha\beta}$ similar to the ReLU case in Proposition 3.4, however in this case $\rho_t^{\alpha\beta}$ is not necessarily equal to 1. At the same time, the linear network limit in case (iii) also has the same covariance SDE as Proposition 3.4.

## 4 Consequences, Discussion, and Future Directions

So far, we have derived the Neural Covariance SDE. Analysis of this SDE reveals important behaviour of the network on initialization. Here we lay out one concrete example and provide some discussion and future directions.

**Exploding and Vanishing Norms.** Here we consider the behaviour of shaping smooth activation functions, as it is done in the experiments of [38]. While the authors here avoided exploding and vanishing norms by numerically optimizing shaping parameters, we can actually describe the precise behaviour a priori with the Neural Covariance SDE. Recall the shaping parameter $a$ from Definition 3.6. Let $V_t$ be the solution to the SDE in (3.9). We can write down the marginal SDE for $V_t^{\alpha\alpha}$ as

$$dV_t^{\alpha\alpha} = \left(\frac{3}{4}\varphi''(0)^2 + \varphi'''(0)\right)\frac{V_t^{\alpha\alpha}}{a^2}(V_t^{\alpha\alpha} - 1)\, dt + \sqrt{2}V_t^{\alpha\alpha}\, dB_t, \quad (4.1)$$

which implies by Proposition 3.7 that $V_t$ has a finite time explosion (with non-zero probability) **if and only if** $\frac{3}{4}\varphi''(0)^2 + \varphi'''(0) > 0$. This criterion can be used to help choose how activation functions should be centered for shaping; below are two examples.

**Example 4.1** (Sigmoid and $\tanh$ at $x_0 = 0$)**.** *We start with the sigmoid activation* $\sigma(x) = \frac{1}{1+e^{-x}}$, *then we can define* $\varphi(x) := 4\sigma(x) - 2$ *to satisfy Assumption 3.5, which leads to* $\varphi''(0) = 0, \varphi'''(0) = -\frac{1}{2}$, *and therefore leads to a stable network. It turns out* $\varphi(x) := \tanh(x)$ *already satisfies Assumption 3.5, which leads to* $\varphi''(0) = 0, \varphi'''(0) = -2$, *and therefore is also stable.*

More generally, if $\sigma$ behaves like a cumulative distribution function for a symmetric unimodal density, we will have that $\varphi''(0) = 0$ and $\varphi'''(0) < 0$ as desired.

**Example 4.2** (Soft Plus at General $x_0 \in \mathbb{R}$). *Let us consider $x_0 \in \mathbb{R}$ and $\sigma(x) = \log(1 + e^{x+x_0})$, which implies $\varphi(x) := (1 + e^{-x_0})\log\frac{1+e^{x+x_0}}{1+e^{x_0}}$ satisfies Assumption 3.5. This gives us $\varphi''(0) = \frac{1}{1+e^{x_0}}, \varphi'''(0) = \frac{1-e^{x_0}}{(1+e^{x_0})^2}$, and therefore $\frac{3}{4}\varphi''(0)^2 + \varphi'''(0) = \frac{1}{(1+e^{x_0})^2}\left(\frac{5}{4} - e^{x_0}\right)$. In other words, the shaped network is stable if and only if $x_0 \geq \log\frac{5}{4}$ (see Figure 2). We note that the authors of [38] numerically found a shift of $x_0 \approx 0.41$, which is in the stable regime of $x_0 \geq \log\frac{5}{4} \approx 0.097$.*

**Relationship to Edge of Chaos.** The finite time explosion example above resembles the Edge of Chaos (EOC) analysis of gradient stability [43, 35, 44, 45], where the weight and bias variance at initialization determines a stability criterion. However, we note that the EOC regime is sufficiently different that the results are not directly comparable. More precisely, the EOC analysis is in the sequential limit of infinite-width and then infinite-depth, which also leaves the activation function unchanged. Under very weak assumptions, the variance (diagonal of $V_t$) will not explode in this regime; instead, the gradient can explode due to the covariance (off diagonals). On the other hand, our finite explosion result is in the joint limit of depth and width, where the variance (diagonal of $V_t$) can explode instead.

**Posterior Inference.** Similar to the NNGP setting, we can use the Neural Covariance SDE to generate a prior over functions $f : \mathbb{R}^{n_{\text{in}}} \to \mathbb{R}^{n_{\text{out}}}$. Consequently, an interesting future direction would be to study the posterior distribution, i.e. the output $z_{\text{out}}^{m+1}$ conditioned on $x^{m+1}$ and a training dataset $(x^\alpha, z_{\text{out}}^\alpha)_{\alpha=1}^m$. However, to our best knowledge, it is not straightforward to explicitly compute or sample from the conditional distributions for this SDE structure. It would be desirable to extend existing approaches in the perturbative regime [30, 31] to our setting.

**Extension to Other Architectures.** The key step to deriving the covariance SDE is the conditional Gaussian distribution in (2.7), which directly leads to a Markov chain. It follows immediately that ResNets [46] admit a similar conditional structure. With a bit more work for convolutional networks, we can obtain $z_{\ell+1}^\alpha | \mathcal{F}_\ell \sim \mathcal{N}(0, \mathcal{A}(V_\ell) \otimes I_n)$ where $\mathcal{A}$ is an affine transformation and $V_\ell$ is the previous layer's Gram matrix [47]. We note that recurrent networks will not lead to a Markov chain or SDE limit, as the weight matrix is reused from layer to layer.

**Simulating SDEs.** Both the Markov chains and SDEs predict neural networks at initialization very well (see Figure 1), but the SDE is significantly faster to simulate. In particular, we can view the Markov chain as an approximate Euler discretization of the SDE, but with a very small step size $n^{-1}$. In contrast, to simulate the SDE we should only need a step size that is small on the scale of depth-to-width ratio $T = d/n$, which is *independent of width $n$*. Therefore, practitioners using the shaping techniques of [38, 39] can now simulate the covariance SDEs at a low computational cost to significantly improve estimates of the output correlation (see Figure 1 and additional simulations in Appendix F).

**Analytical Tractability of SDEs.** Besides numerical tractability, the SDEs are also far more tractable to analyze. For example, in the one input case, we arrive at geometric Brownian motion (2.6), which is known to have a log-normal distribution at fixed times. Similarly, our finite time explosions hinge on the fact we identified an SDE limit. In the same way that NNGP theory played a major role in the infinite-width regime, the Neural Covariance SDEs and the techniques developed here also serve as a mathematical foundation for studying training and generalization.

## Acknowledgement

We would like to thank Sinho Chewi, James Foster, Boris Hanin, Cameron Jakub, Jeffrey Negrea, Nuri Mert Vural, Guodong Zhang, Matthew S. Zhang, and Yuchong Zhang for helpful discussions and draft feedback. We would like to thank Sam Buchanan and Soufiane Hayou for pointing out a gap in the proof of Proposition B.8. ML is supported by Ontario Graduate Scholarship and the Vector Institute. MN is supported by an NSERC Discovery Grant. DMR is supported in part by Canada CIFAR AI Chair funding through the Vector Institute, an NSERC Discovery Grant, Ontario Early Researcher Award, a stipend provided by the Charles Simonyi Endowment, and a New Frontiers in Research Exploration Grant.

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
