# OpenReview forum: "The Neural Covariance SDE: Shaped Infinite Depth-and-Width Networks at Initialization"
_NeurIPS.cc/2022/Conference — NeurIPS 2022 Accept_

### Official Review · Reviewer_TjDp · 2022-07-09

**Rating:** 8
**Confidence:** 2
**Soundness:** 4 excellent
**Presentation:** 3 good
**Contribution:** 4 excellent

**Summary:**

This paper is theoretical and studies the behavior of neural networks in the shaped infinite depth/width limit. As stated in the appendix, the logits at the end of a feedforward network are conditionally Gaussian at initialization with a random covariance matrix. This work studies that matrix at infinite depth/width.

Current theory does not work at large depths because small fluctuations between layers are disregarded. When properly accounted for, this paper demonstrates that the fluctuations add up so that the covariance matrix is governed by a stochastic differential equation (SDE). This work therefore provides a stronger theoretical understanding about networks at initialization.

The distribution of the covariance matrix closely follows the distribution predicted using the SDE theory developed here, this is shown by simulating the initialization of many neural networks.

**Questions:**

- Is there a way this work can be used outside of our understanding? For example to find a valuable initialization scheme?
- Is it possible to extend this to understand training dynamics?

**Limitations:**

The societal impact is not addressed. However, this is a theoretical paper and a broader impact statement is not required (though it is always appreciated).

The limitations are discussed in the introduction. Briefly, the paper only applies to initialization and not training dynamics.

**Strengths And Weaknesses:**

Strengths:

The paper provides a creative theoretical contribution. Furthering our understanding of large neural networks, from a novel angle using SDEs. The work is also tested briefly, leading to positive results like those in Figure 1.

Weaknesses:

The only weakness I can find is in the presentation, which is mainly down to personal preference. In my opinion the paper feels very math heavy, and possibly not as accessible as it could be. The paper may benefit from writing more intuitive statements in the main paper and what their consequences are, essentially walking through the work. Then the more rigorous proofs and statements can be placed in the appendix. For example, equation 2 for the ReLU-like activations is currently:

$\phi(x) = s_{+}\max(x, 0) + s_{-}\min(x, 0) = s_{+}\text{ReLU}(x) - s_{-}\text{ReLU}(-x)$

These are both technically correct, but an easier way of writing may be:

$\phi(x) = s_{+}x ~~ \text{if} ~~ x\geq 0,
~~ \text{or} ~~
s_{-}x ~~ \text{if} ~~ x< 0$

Additionally more figures/tables showing visually how well the Covariance SDE works always help.

Minor Typos (do not affect the quality of the paper):

- Line 60: "matches matches" should be "matches"
- Line 246: "Not only do these stopping time rules out" should be "Not only do these stopping times rule out" or "Not only does this stopping time rule out"

---

> ### Author Response · Authors · 2022-08-01
> **Response to Reviewer TjDp**
>
> We want to thank you for the generous comments and score. We agree parts of the current draft are quite math heavy, and we will incorporate more helpful intuitive statements into the next iteration of our draft.
>
> We will also briefly address your questions below.
>
> # Applications of Our Results
>
> The most direct application of our results is towards finding a good shaping scheme for training. Our work is strongly motivated by the extensive training experiments from [1,2], where the authors first showed that shaping activation functions can significantly improve the speed of training, even without normalization layers. With our SDEs, we can improve on the infinite-width analysis and tuning of the shaping parameters, and this can translate to even better training performance.
>
> # Understanding Training Dynamics
>
> Indeed, we hope this work serves as the first towards analyzing training dynamics. As NNGP established the technical steps towards analyzing the NTK, we hope that the proof techniques for deriving the covariance SDE can also derive a similar SDE for the NTK in the infinite-depth-and-width limit. From this point, we hope that in the future, we can say more about the properties of training neural networks in this regime.
>
> # References
>
> 1.  Martens, J., Ballard, A., Desjardins, G., Swirszcz, G., Dalibard, V., Sohl-Dickstein, J. and Schoenholz, S.S., 2021. Rapid training of deep neural networks without skip connections or normalization layers using deep kernel shaping. arXiv preprint arXiv:2110.01765. https://arxiv.org/pdf/2110.01765.pdf
>
> 2. Zhang, G., Botev, A. and Martens, J., 2022. Deep Learning without Shortcuts: Shaping the Kernel with Tailored Rectifiers. arXiv preprint arXiv:2203.08120. https://arxiv.org/pdf/2203.08120.pdf

---

> > ### Comment · Reviewer_TjDp · 2022-08-04
> > **Thank you for your response**
> >
> > Dear authors,
> >
> > Thank you for your response. This answers my questions. My score remains the same.

---

### Official Review · Reviewer_SK2F · 2022-07-09

**Rating:** 6
**Confidence:** 3
**Soundness:** 3 good
**Presentation:** 2 fair
**Contribution:** 3 good

**Summary:**

This paper study the random covariance matrix in the shaped infinite-depth-and-width limit to model the layer to layer fluctuations in random feature neural networks.

The authors claim that the modeling of covariance and depth has a significant improvement over the NNGP regime.

**Questions:**

Is there any further interpretation when incorporating this technique to neural tangent kernel?


**Limitations:**

The analysis is still in the linearization regime, which is known underestimate the power of neural networks. The evidence is not strong enough to justify the necessity of “reshaping” in NN literature at this moment.

**Strengths And Weaknesses:**

Strength

The idea of depth and covariance modeling is intuitive and non-trivial. This paper analyzes infinite-depth-and-width limit by reshaping the activation function, resulting non-trivial SDE structure.

Weakness

Although the “reshaping” idea is theoretically elegant, the practical interpretation is still based on the linearization regime, which is known to underestimate the neural networks.

Besides, the gain from “reshaping” is not significant enough to replace NNGP/NTK (at this moment).
The theoretical modeling is based on the evolution of random features with the depth. However, deep random features are not better than two-layer one in real practice. More interpretation might be needed to justify the significance.

The authors raise some examples such as the Exploding and Vanishing Norms. However, it is an artificial setting which is not even a problem in real practice. A more practical example indicated by this framework is appreciated.

---

> ### Author Response · Authors · 2022-08-01
> **Response to Reviewer SK2F - Part 1**
>
> We want to thank you for your careful comments and questions.
>
> On a high level, we believe you have the impression that we are studying a model of only theoretical interest and no connection to real networks. (Indeed, you have the impression that the model is even "in the linear regime".) The entire point of this work is to identify a model that matches deep learning practice better than the infinite-width model, and so this impression---that we are working on a model that is disconnected from practice---is indeed problematic!
>
> Below, we expand on the following key points:
> 1. There's a growing body of evidence that the infinite-depth-and-width model better matches the properties of real networks at initialization than the infinite-width model.
> 2. As networks with standard initializations get deeper, training usually suffers unless batch normalization or similar techniques are used. Shaping has recently emerged as a technique for designing deep networks that achieve the same accuracy but can be trained without batch norm. Thus, shaped infinite-depth-and-width limits are an attempt at a model of this new class of networks.
> 3. Among the shaped infinite-depth-and-width limits, one can indeed design the amount of leakiness in order to arrive at a "linear limit" corresponding to infinitely deep linear networks. However, we have also uncovered the precise amount of shaping to arrive at a nonlinear limit and an SDE describing the covariance.
>
> We think readers would benefit from knowing just how promising shaped neural networks and the infinite-depth-and-width limit are for arriving at a tractable model corresponding to practical high-performance networks. We'll incorporate key parts of this evidence into our updated draft.
>
> # Infinite-Depth-and-Width Limit and Empirical Evidence
>
> In [3,4], they study the unshaped model (in completely connected and ResNet architectures) and show that the activations are log-Gaussian. This matches observations made in recent empirical work [5]. In [4], they even compare the precise log-Gaussian limit for activations in deep ResNets at initialization to those of standard (fixed depth and width) ResNets architectures and observe a nearly perfect match. In contrast, [4] show that the infinite width model's prediction for activations at initialization do not match.
>
> # Practical Impacts: Shaping Significantly Improves Training
>
> We agree that we can improve the discussion towards the practical impacts of our work, and in particular around the method of shaping activation functions. Our work is strongly motivated by the extensive experiments by Martens et al. [1] and Zhang et al. [2], where the authors first showed that shaping activations significantly improves the speed of training. Furthermore, the same work also showed that with activation shaping, normalization layers are no longer required at the size of standard ResNets trained on CIFAR-10. See section 12.4 of [1] for a more detailed discussion on the issues arising from a degenerate (unshaped) network. (We do also note [1] considered the shaping of softplus activations, which would be captured by our exploding and vanishing norm result.)
>
> However, the analyses of [1,2] are limited to the infinite-width setting, and therefore cannot capture the randomness of covariance (and correlation) throughout the layers. In particular, by observing the bottom right plot of Figure 1, we can see that the distribution of correlation $\rho^{\alpha\beta}_d$ is *far closer to being degenerate* (close to $1$) than the infinite-width theory predicts. This implies that to properly apply the shaping method, the practitioner *must account for the randomness* introduced by the layerwise propagation.
>
> The covariance SDE we introduce is also cheap to simulate, as discretizing SDEs do not require a vanishingly small step size regardless of depth of the network. This is in comparison to the existing methods of [1,2], where authors here need to compute a recursion through the number of layers.
>
> To summarize, shaping is a proven method that improves training, and our theory helps improve the application of this method. We will add the above discussion on practical impacts to the updated draft, as much as the space allows.

---

> ### Author Response · Authors · 2022-08-01
> **Response to Reviewer SK2F - Part 2**
>
> # Theoretical Impacts: Beyond the Linear Regime
>
> We agree with you that the linear regime significantly underestimates the capabilities of real neural networks, which is the main motivation for our work. In particular, our analysis is *not in the linear regime*, and it actually matches the behaviour of *finite size networks*.
>
> We believe you have mistaken the evolution of the random covariance matrix as the same as stacking random features (we will clarify this further in the next draft). In particular, we have empirically verified the infinite-depth-and-width limit matches real networks at initialization (see Figure 1 and the Appendix for additional simulations). Furthermore, it has been shown that there is feature learning in this regime [3], which is not possible for linear models. We believe studying initialization is an important first step, as the work on NNGP proved to pave the way for NTK analysis to follow.
>
> Additionally, as opposed to infinite-width (finite-depth) networks and random feature models, our limit cannot be characterized by a single fixed kernel. Perhaps the easiest way to see this is by observing that the diffusion coefficient $\Sigma$ in equation 12 is dense, which means the evolution of a single covariance entry $V^{\alpha\beta}_t$ depends on all the other covariances. In other words, the covariance for inputs $x^1, x^2$ will depend on all the other input data points $x^3, x^4, \cdots, x^m$!
>
> To summarize, we do indeed study a limit of neural networks outside the linear regime, and we will further clarify this fact in the paper.
>
> # Future Work: Neural Tangent Kernel
>
> To address your question with regards to the neural tangent, indeed we are actively thinking about extending this to the NTK. As [3] have shown, the NTK object will be a random and time-evolving object (i.e. there is feature learning). We believe that our techniques can extend towards this direction, and hopefully we can study neural networks beyond initialization.
>
> # References
>
> 1.  Martens, J., Ballard, A., Desjardins, G., Swirszcz, G., Dalibard, V., Sohl-Dickstein, J. and Schoenholz, S.S., 2021. Rapid training of deep neural networks without skip connections or normalization layers using deep kernel shaping. arXiv preprint arXiv:2110.01765. https://arxiv.org/pdf/2110.01765.pdf
>
> 2. Zhang, G., Botev, A. and Martens, J., 2022. Deep Learning without Shortcuts: Shaping the Kernel with Tailored Rectifiers. arXiv preprint arXiv:2203.08120. https://arxiv.org/pdf/2203.08120.pdf
>
> 3. Hanin, B. and Nica, M., 2019. Finite depth and width corrections to the neural tangent kernel. arXiv preprint arXiv:1909.05989. https://arxiv.org/pdf/1909.05989.pdf
>
> 4. Li, M., Nica, M., Roy, D.M. 2020. The future is log-Gaussian: ResNets and their infinite width and depth limit at initialization. Advances in Neural Information Process Systems.
>
> 5. Chmiel, B., Ben-Uri, L., Shkolnik, M., Hoffer, E., Banner, R. and Soudry, D., 2021. Neural gradients are near-lognormal: improved quantized and sparse training. In Proc. of Int Conf. Learning Representations (ICLR) https://openreview.net/forum?id=EoFNy62JGd

---

> > ### Comment · Reviewer_SK2F · 2022-08-04
> > **Response**
> >
> > We appreciate the reviewers' great clarification and that resolve some of my concerns. I have raise my scores but I do think it is still necesary to rearrange the contents to emphasize the significance of this work.

---

### Official Review · Reviewer_xxTF · 2022-07-25

**Rating:** 8
**Confidence:** 4
**Soundness:** 3 good
**Presentation:** 4 excellent
**Contribution:** 3 good

**Summary:**

Infinite-depth-and-width limits for the covariances of activation layers are derived for neural networks with ReLU-like and smooth activation functions at initialization (under the iid Gaussian weights assumption). To obtain a nontrivial limit, leaky ReLU-like activations are assumed with the slopes differing by a factor decreasing as $O(n^{-1/2})$. This factor is shown to be critical for this limit. For smooth activation functions, a criterion is obtained on the activation function to ensure that the limiting covariance SDE is nonexplosive. Sigmoidal, tanh, and some classes of softplus activation functions are determined to satisfy this criterion. Some experiments are conducted which show agreement between the distribution of correlations for initialized finite NNs and those of the SDE predictions.

**Questions:**

How does the kernel derived from these neural covariance SDEs behave in a GP relative to e.g. the arc-cosine kernel?

Does the typical ReLU setup $s_+ = 1$, $s_- = 0$ not reveal an interesting limiting SDE? This seems like a surprising omission, given that almost any other activation function seems to exhibit nontrivial limiting behaviour.

**Limitations:**

While the authors do not discuss limitations of their own work, I find this to be reasonable, as there are few apparent limitations here. Perhaps the most significant is the focus on initialized networks (requiring all weights to be iid normal), which has surprisingly little discussion in the main document.

**Strengths And Weaknesses:**

NNGPs and NTKs are among the most useful theoretical tools for analyzing neural networks, and provide a powerful link between the versatile Gaussian process paradigm and deep learning. However, it has been shown time and again that these infinite-width approximations do not mimic the performance of modern neural networks. Here, the authors have instead considered the infinite-depth-and-width regime, and while analogues of NNGPs and NTKs are not yet available, the proposed building blocks are provided here for neural networks at initialization. Comfortingly, the behaviour of these approximations seem to mimic those of finite NNs --- as in double descent theory, the large n,d regime appears to be the correct one. The theoretical analysis is extensive, and the presentation is (mostly) excellent, although I did not have the opportunity to inspect the proofs in detail.

With this in mind, and given that this work is novel to my knowledge, I believe this is a valuable contribution and am excited by potential follow-up work here. However, on first read, I was having a difficult time drawing conclusions from these particular results. In particular, I believe the numerical results and their details deserve their own section in the main document. To make room for this, I would suggest shrinking Section 2.3 (I found this section to be more confusing than enlightening, although I understand why it is here), and moving the critical exponent results to the Supplementary Material, as I believe they can be easily summarized in a sentence or two. It would also have been nice to see what these covariances could do as part of a GP, but I would not expect any significant improvements over known kernels.

Some minor points:
- The first paragraph of the abstract is confusing as written; I would strongly suggest rewriting it, starting with 'Recent work...' in the first sentence
- line 72: 'according a' -> 'according to a'
- line 90: 'chains' should probably be capitalized to be consistent
- line 103: 'of, the' -> 'of the'
- line 106: 'Gaussian random matrix' - Define. There should also be explicit mention of the Gaussian weights assumption here.
- line 108: Consider reversing the order of these two equations, as the significance of g is unclear when reading from left-to-right.
- line 119-121: last sentence is unnecessary
- line 161: replace colon with full stop
- line 162-163: last sentence is repetitive; consider removing
- Can different symbols be used to describe the Markov chain and the limiting SDE (e.g. \mathcal{V})? The results are difficult to parse as is.
- line 187: 'they are' -> 'they follow'
- line 232: I'm not sure the exclamation mark is warranted here...
- line 274: Proposition 7 should probably be moved here to avoid confusion

---

> ### Author Response · Authors · 2022-08-01
> **Response to Reviewer xxTF**
>
> Thank you for your generous comments and score. We are happy that you share our excitement around potential follow-up work. While this work is focused on initialization, the SDEs we derive should factor into the subsequent study of training.
>
> # Unshaped ReLU networks
>
> We wanted to address your question on the limit for unshaped ReLU networks. This is actually the motivation for section 2.3. The Markov chain in eq. 10 is not of the type
>
> $$Y_{\ell+1} = Y_\ell + \frac{b(Y_\ell)}{n} + \frac{\sigma(Y_\ell)}{\sqrt{n}} \xi_\ell + O(n^{-3/2}) \,,
> $$
>
> because it has the form $Y_{\ell+1} = f(Y_\ell) + \cdots$.  For there to be an SDE limit, the difference between $Y_\ell$ and $Y_{\ell+1}$ must vanish as $n\to\infty$, as this corresponds to $X_t$ and “$X_{t+dt}$”. In fact, the limit is degenerate in the sense that the correlation being all pairs of points converges to 1. In other words, the network function at initialization is a.s. constant (over different inputs), even if the value of the constant is random.
>
> Subsequent to finding this degenerate limit, we discovered the empirically promising results for shaping and recognized that the shaped Markov chains lead to SDE limits with more promising behavior.
>
> # Other detailed suggestions
>
> We appreciate all the detailed suggestions you have, and we will incorporate them in the next iteration of our draft.

---

> > ### Comment · Reviewer_xxTF · 2022-08-08
> > **Thank you for your response**
> >
> > Thank you for responding to my comments and for the clarification regarding unshaped ReLU. I have revisited this paper a number of times since this review, and have come to appreciate it more each time. Consequently, I have decided to raise my score.
> >
> > A few more suggestions: I believe that moving parts of Appendix E into the main body of the paper using the additional allowed page could help to make the presentation less dense for the reader. In line with Reviewer stwd, I agree that including a few sentences on possible approaches for inference (similar to what is discussed in your response) would be useful. I would also recommend adding some paragraph headings to Section 2.3 to break up the text (e.g. covariance / correlation).

---

### Official Review · Reviewer_stwd · 2022-07-26

**Rating:** 7
**Confidence:** 3
**Soundness:** 4 excellent
**Presentation:** 3 good
**Contribution:** 3 good

**Summary:**

The paper studies the random covariance matrix of infinite depth and width limit of initialized deep neural networks. Using recent work on deep kernel shaping, the authors show that shaping activation function leads to non-trivial stochastic differential equations (SDE) of neural covariance matrix. Authors show that simulation of finite networks match the distribution predicted by the SDE.

**Questions:**

One limitation I see from the current approach is how to use the current framework to do **inference** In the conventional infinite-width limit, due to deterministic covariance matrix inference leads to conventional Gaussian process inference. With non-deterministic covariances, does the author have ways to perform posterior inference to make predictions? Interesting related work is [Yaida, Non-Gaussian processes and neural networks at finite widths, MSML 2020] (also covered in 6.4 of ref [30] Roberts, Yaida and Hanin 2022) which looks at finite width correction to NNGP and obtain non Gaussian process as correction to the infinite-width NNGP but still retains ways to perform inference as a correction to NNGP inference.

Also for the first attempt of a new theoretical approach, simplicity is a virtue and I appreciate it. However, it would be beneficial for the readers to see applicability / limitation of neural covariance SDE to general architecture. For example, DKS[37] or TAT[38] both main practical applications are to Convolutional Networks, setup in Section 2 deals just with fully connected networks. Could authors expand more on generalization to more structured architecture?

What is the practical benefit of having SDEs vs just Markov Chains (as in section 2.3)? I understand there's a nice "trainability" benefit coming from activation shaping and SDE maybe coming along with the choice. But my read from Figure 1 in terms of predictability of the theory of neural covariances, both are as good matching the simulations.

In the setup, neural networks weights are initialized with 0-mean variance 1 Gaussian without any bias terms. Often exploding/vanishing of input norms are phrased in terms of initialization variances of weights and biases(e.g. Schoenholz et al., Deep Information Propagation, ICLR 2017 or [34]) and denoted Chaotic/Critical/Ordered phase. Wonder if the condition in section 4, eq (24) is directly related to the phases if accounted for weight/bias variances. Or should I consider if and only if condition in (24) as totally different conditions due to simultaneous width-depth limit?

Simple Nit: L200 should be either PDE or derivative on $\rho_t$ should be full derivative? Maybe the point is to say it's not a stochastic DEs? From the first read, it was slightly confusing.


**Limitations:**

Mostly a theoretical analysis of deep neural networks, so there's no foreseen negative societal impact.

Few limitations as raised in the Questions section: (not necessarily needs resolving but would be good if mentioned as future directions etc)
- Methods to perform inferences
- Generalization beyond fully connected networks


**Strengths And Weaknesses:**

The paper presents an interesting theoretical framework for understanding simultaneously the large depth + width limit of random neural networks at initialization. Much of conventional large-width limit has focused on fixed depth and large width limit but there is evidence as presented by the authors as well that if one simultaneously takes large width and large depth limit, the behavior of neural networks can be systematically different.

Interesting contribution of this paper is that even in this seemingly complicated limit, there's tractable theory determining the random covariance matrix in terms of stochastic differential equations. This is allowed by utilizing activation shaping which was used to show that one could train normalization-layer and skip-connection free very deep feed-forward networks.

Paper does a very thorough job of covering related works, with the only exception I see being [Yaida, Non-Gaussian processes and neural networks at finite widths, MSML 2020] mentioned in the Questions section.

Exposition is clear. The exposition split of starting from unshaped ReLU networks in Section 2 with SDE example in section 2.2 then leading to shaped activation case of Section 3 was quite helpful.  However, to understand the full details of the work, it does require readers to have some prior exposure to modern probability and stochastic processes. Appendix A attempts to provide some background but this serves more of a refresher than an introduction. This may limit the target audience. Having said that, probably simple exposure to stochastic processes and stochastic differential equations may be sufficient to understand the main paper's message.

Authors did a great job of providing, ipython notebook of the empirical simulations. Empirical results do show solid support for correctness and generality of derived results. One thing to be cautious about is to consider removing unnecessary outputs in the notebook, since there is some identifying information left by mistake. This doesn't seem to be a severe violation of anonymity but needs to be careful for future submissions.

---

> ### Author Response · Authors · 2022-08-01
> **Response to Reviewer stwd - Part 1**
>
> We want to thank you for the generous comments and score. We are happy to see that you enjoyed reading our work. You have raised some good questions, and we hope to provide satisfying answers to the best of our abilities below.
>
> # Posterior Inference
>
> Posterior inference with conditionally Gaussian processes is a challenging problem, even when the kernel lives in a parametric family. Now that the kernel is the output of an SDE (which we could view as a prior), the problem is even more challenging. Prior work (that you have highlighted) may point to some approaches. Another approach is to consider MCMC or variational inference over the joint model of the Gram matrix and GP outputs. It seems feasible for a small number of data points.
>
> There is interest in this SDE beyond inference however. Like how the NTK is defined in terms of the NNGP kernel in the infinite-width theory, we might hope to find an analogous relationship in this shaped-infinite-depth-and-width theory. This would yield a theory of training dynamics near initialization.
>
> # Extending to Convolutional Networks
>
> We agree that it will be interesting to understand the role of the architecture in the Neural Covariance SDE. To increase width for a convolutional neural network, the standard approach is to increase the number of channels [1]. Recall in our derivation, we needed to identify the covariance structure from multiplying a Gaussian matrix in equation 7; for convolution with a random filter, this leads to the conditional covariance structure in equation 4 of [1]. The authors of [1] called the corresponding matrix $\mathcal{A}(K^l)$, which is the same as our $V_\ell$. It remains to Taylor expand $V_{\ell+1} | V_\ell$ with respect to $n^{-1/2}$ to recover the Markov chain and SDE.
>
> In summary, we believe our results should extend to convolutional architectures, albeit the detailed calculations may yield some new difficulties that we need to handle carefully.
>
> # On the Practical Benefit of SDEs
>
> Both the Markov chains and SDEs predict neural networks at initialization very well.
>
> One way to see the advantage of the SDE over the Markov chain is to view the Markov chain as an Euler discretization of the SDE, but with a possibly very small step size $n^{-1}$. In contrast, to simulate the SDE we should only need a step size that is small on the scale of depth-to-width ratio $T = d/n$, which is independent of width $n$!
>
> Besides numerical tractability, there is analytical tractability. Noticing that one has found a Markov chain with an SDE limit can often lead to considerable simplification. For example, in the one input case, we arrive at geometric Brownian motion (equation 6), which is known to have a log-normal distribution at fixed times. The Markov chain, with all terms of all orders, obscures this. By discarding terms of sufficiently low-order (i.e., by studying the SDE limit!), we find this simpler structure.
>
> Similarly, our finite time explosions hinge on the fact we identified an SDE limit. In the future, we also hope to recover additional closed form expressions governing the behaviour of the covariance SDE, so that we can say more about the solution’s properties.
>
> Overall, our goal of this work was to take the first step towards a more tractable theory of neural networks in the infinite-depth-and-width regime, similar to the way NNGP results pointed the way to other infinite-width results, such as the NTK.

---

> > ### Comment · Reviewer_stwd · 2022-08-08
> > **Thank you for the response**
> >
> > Thank you for clarifying all my questions. I believe including these clarifying discussions into the main text will improve the paper for the NeurIPS readers.
> >
> > My score remains the same and I am happy to champion the paper for acceptance.

---

> ### Author Response · Authors · 2022-08-01
> **Response to Reviewer stwd - Part 2**
>
> # Relationship to Chaotic/Ordered Phases
>
> There’s definitely a similar phase transition behaviour to the chaotic/ordered regimes of [2,3], but as you suspected, our setting is different enough to warrant a more careful comparison. We hope to do a more careful analysis of the gradient in the future, but for now, we will provide a high level discussion.
>
> The authors of [2,3] considered a bounded activation function in the infinite-width limit with large depth, in which case their variance $V^{\alpha\alpha}_\ell$ always converged to a finite fixed point when depth is large [2, eq. 3]. Their chaotic and ordered phases are then defined by the behaviour of the correlation fixed point [2, eq. 5], which in turn determines the behaviour of the gradient [2, eq. 16].
>
> In our case, shaping the activation leads to an unbounded function, and consequently the variance $V^{\alpha\alpha}_\ell$ is not always bounded - even if we take the same limit as [2] (but shaping depends on depth instead like the DKS/TAT papers), in which case we get an ODE with finite time explosion. Intuitively, if we drop the Brownian motion from eq. 18 and consider
>
> $$ dX_t = b X_t (X_t - 1) \, dt \,, $$
>
> which is the logistic ODE, and has a finite time explosion if $X_0 > 1$ and $b > 0$. At the same time, due to shaping, our correlation $\rho^{\alpha\beta}_t$ will actually be able to avoid the fixed point (i.e., non-degenerate). So the gradient will be well behaved from the perspective of correlations (we are in the critical regime defined by [2]), but it may still explode due to variances exploding.
>
> # Minor Points
>
> We will add the suggested reference, and indeed for line 200 we meant that it is simply an ODE. We will modify the notation to clarify this point. We've also done a pass through the notebook, cleaning up file paths. Perhaps it makes sense to also remove them from the review.
>
> # References
>
> 1. Novak, R., Xiao, L., Lee, J., Bahri, Y., Yang, G., Hron, J., Abolafia, D.A., Pennington, J. and Sohl-Dickstein, J., 2018. Bayesian deep convolutional networks with many channels are gaussian processes. arXiv preprint arXiv:1810.05148. https://arxiv.org/pdf/1810.05148.pdf
>
> 2. Schoenholz, S.S., Gilmer, J., Ganguli, S. and Sohl-Dickstein, J., 2016. Deep information propagation. arXiv preprint arXiv:1611.01232. https://arxiv.org/pdf/1611.01232.pdf
>
> 3. Yang, G. and Schoenholz, S., 2017. Mean field residual networks: On the edge of chaos. Advances in neural information processing systems, 30. https://arxiv.org/pdf/1712.08969.pdf

---

### Meta-Review · Area_Chair_iVHV · 2022-08-26

**Recommendation:** Accept
**Confidence:** Certain

**Metareview:**

There is a clear consensus to accept this manuscript.  The results are impressive, and have a nice theoretical orientation that will allow the results to have continued impact as the field advances.  There are some minor errors by the authors in the discussions, which are worth the authors being aware of before submitting their final version.  In particular, they state that:

"The authors of [2,3] considered a bounded activation function in the infinite-width limit with large depth, in which case their variance
V
ℓ
α
α
 always converged to a finite fixed point when depth is large [2, eq. 3]. Their chaotic and ordered phases are then defined by the behaviour of the correlation fixed point [2, eq. 5], which in turn determines the behaviour of the gradient [2, eq. 16].

In our case, shaping the activation leads to an unbounded function, and consequently the variance
V
ℓ
α
α
 is not always bounded - even if we take the same limit as [2] (but shaping depends on depth instead like the DKS/TAT papers), in which case we get an ODE with finite time explosion. Intuitively, if we drop the Brownian motion from eq. 18 and consider

d
X
t
=
b
X
t
(
X
t
−
1
)
,
d
t
,
,

which is the logistic ODE, and has a finite time explosion if
X
0
>
1
 and
b
>
0
. At the same time, due to shaping, our correlation
ρ
t
α
β
 will actually be able to avoid the fixed point (i.e., non-degenerate). So the gradient will be well behaved from the perspective of correlations (we are in the critical regime defined by [2]), but it may still explode due to variances exploding.

References
Novak, R., Xiao, L., Lee, J., Bahri, Y., Yang, G., Hron, J., Abolafia, D.A., Pennington, J. and Sohl-Dickstein, J., 2018. Bayesian deep convolutional networks with many channels are gaussian processes. arXiv preprint arXiv:1810.05148. https://arxiv.org/pdf/1810.05148.pdf
Schoenholz, S.S., Gilmer, J., Ganguli, S. and Sohl-Dickstein, J., 2016. Deep information propagation. arXiv preprint arXiv:1611.01232. https://arxiv.org/pdf/1611.01232.pdf
Yang, G. and Schoenholz, S., 2017. Mean field residual networks: On the edge of chaos. Advances in neural information processing systems, 30. https://arxiv.org/pdf/1712.08969.pdf
"

And while [2] states they consider bounded activations, it is not used or necessary and is not used in [3] or subsequent more recent work that discusses the edge of chaos further; see for instance: Activation function design for deep networks: linearity and effective initialisation by Murray et al. and On the impact of the activation function on deep neural networks training by Hayou et al.


**Award:**

Yes

---

### Decision · Program_Chairs · 2022-09-14

Accept